# Assessing the effect of the COVID-19 pandemic, shift to online learning, and social media use on the mental health of college students in the Philippines: A mixed-method study protocol

**Leonard Thomas S. Lim**[1], **Zypher Jude G. Regencia**[2,3], **J. Rem C. Dela Cruz**[1], **Frances Dominique V. Ho**[1], **Marcela S. Rodolfo**[1], **Josefina Ly-Uson**[4], **Emmanuel S. Baja**[2,3]*

**1** College of Medicine, University of the Philippines, Manila, Philippines, **2** Department of Clinical Epidemiology, College of Medicine, University of the Philippines, Manila, Philippines, **3** Institute of Clinical Epidemiology, National Institutes of Health, University of the Philippines, Manila, Philippines, **4** Department of Psychiatry, College of Medicine, University of the Philippines, Manila, Philippines

* esbaja@up.edu.ph

**Funding:** This project is being supported by the American Red Cross through the Philippine Red Cross and Red Cross Youth. The funder will not

## Abstract

### Introduction

The COVID-19 pandemic declared by the WHO has affected many countries rendering everyday lives halted. In the Philippines, the lockdown quarantine protocols have shifted the traditional college classes to online. The abrupt transition to online classes may bring psychological effects to college students due to continuous isolation and lack of interaction with fellow students and teachers. Our study aims to assess Filipino college students' mental health status and to estimate the effect of the COVID-19 pandemic, the shift to online learning, and social media use on mental health. In addition, facilitators or stressors that modified the mental health status of the college students during the COVID-19 pandemic, quarantine, and subsequent shift to online learning will be investigated.

### Methods and analysis

Mixed-method study design will be used, which will involve: (1) an online survey to 2,100 college students across the Philippines; and (2) randomly selected 20–40 key informant interviews (KIIs). Online self-administered questionnaire (SAQ) including Depression, Anxiety, and Stress Scale (DASS-21) and Brief-COPE will be used. Moreover, socio-demographic factors, social media usage, shift to online learning factors, family history of mental health and COVID-19, and other factors that could affect mental health will also be included in the SAQ. KIIs will explore factors affecting the student's mental health, behaviors, coping mechanism, current stressors, and other emotional reactions to these stressors. Associations between mental health outcomes and possible risk factors will be estimated using generalized linear models, while a thematic approach will be made for the findings from the KIIs. Results of the study will then be triangulated and summarized.

have a role in the study design, data collection and analysis, decision to publish, or preparation of the manuscript.

**Competing interests:** The authors have declared that no competing interests exist.

## Ethics and dissemination

Our study has been approved by the University of the Philippines Manila Research Ethics Board (UPMREB 2021-099-01). The results will be actively disseminated through conference presentations, peer-reviewed journals, social media, print and broadcast media, and various stakeholder activities.

## Introduction

The World Health Organization (WHO) declared the Coronavirus 2019 (COVID-19) outbreak as a global pandemic, and the Philippines is one of the 213 countries affected by the disease [1]. To reduce the virus's transmission, the President imposed an enhanced community quarantine in Luzon, the country's northern and most populous island, on March 16, 2020. This lockdown manifested as curfews, checkpoints, travel restrictions, and suspension of business and school activities [2]. However, as the virus is yet to be curbed, varying quarantine restrictions are implemented across the country. In addition, schools have shifted to online learning, despite financial and psychological concerns [3].

Previous outbreaks such as the swine flu crisis adversely influenced the well-being of affected populations, causing them to develop emotional problems and raising the importance of integrating mental health into medical preparedness for similar disasters [4]. In one study conducted on university students during the swine flu pandemic in 2009, 45% were worried about personally or a family member contracting swine flu, while 10.7% were panicking, feeling depressed, or emotionally disturbed. This study suggests that preventive measures to alleviate distress through health education and promotion are warranted [5].

During the COVID-19 pandemic, researchers worldwide have been churning out studies on its psychological effects on different populations [6–9]. The indirect effects of COVID-19, such as quarantine measures, the infection of family and friends, and the death of loved ones, could worsen the overall mental wellbeing of individuals [6]. Studies from 2020 to 2021 link the pandemic to emotional disturbances among those in quarantine, even going as far as giving vulnerable populations the inclination to commit suicide [7, 8], persistent effect on mood and wellness [9], and depression and anxiety [10].

In the Philippines, a survey of 1,879 respondents measuring the psychological effects of COVID-19 during its early phase in 2020 was released. Results showed that one-fourth of respondents reported moderate-to-severe anxiety, while one-sixth reported moderate-to-severe depression [11]. In addition, other local studies in 2020 examined the mental health of frontline workers such as nurses and physicians—placing emphasis on the importance of psychological support in minimizing anxiety [12, 13].

Since the first wave of the pandemic in 2020, risk factors that could affect specific populations' psychological well-being have been studied [14, 15]. A cohort study on 1,773 COVID-19 hospitalized patients in 2021 found that survivors were mainly troubled with fatigue, muscle weakness, sleep difficulties, and depression or anxiety [16]. Their results usually associate the crisis with fear, anxiety, depression, reduced sleep quality, and distress among the general population.

Moreover, the pandemic also exacerbated the condition of people with pre-existing psychiatric disorders, especially patients that live in high COVID-19 prevalence areas [17]. People suffering from mood and substance use disorders that have been infected with COVID-19 showed higher suicide risks [7, 18]. Furthermore, a study in 2020 cited the following factors

contributing to increased suicide risk: social isolation, fear of contagion, anxiety, uncertainty, chronic stress, and economic difficulties [19].

Globally, multiple studies have shown that mental health disorders among university student populations are prevalent [13, 20–22]. In a 2007 survey of 2,843 undergraduate and graduate students at a large midwestern public university in the United States, the estimated prevalence of any depressive or anxiety disorder was 15.6% and 13.0% for undergraduate and graduate students, respectively [20]. Meanwhile, in a 2013 study of 506 students from 4 public universities in Malaysia, 27.5% and 9.7% had moderate and severe or extremely severe depression, respectively; 34% and 29% had moderate and severe or extremely severe anxiety, respectively [21]. In China, a 2016 meta-analysis aiming to establish the national prevalence of depression among university students analyzed 39 studies from 1995 to 2015; the meta-analysis found that the overall prevalence of depression was 23.8% across all studies that included 32,694 Chinese university students [23].

A college student's mental status may be significantly affected by the successful fulfillment of a student's role. A 2013 study found that acceptable teaching methods can enhance students' satisfaction and academic performance, both linked to their mental health [24]. However, online learning poses multiple challenges to these methods [3]. Furthermore, a 2020 study found that students' mental status is affected by their social support systems, which, in turn, may be jeopardized by the COVID-19 pandemic and the physical limitations it has imposed. Support accessible to a student through social ties to other individuals, groups, and the greater community is a form of social support; university students may draw social support from family, friends, classmates, teachers, and a significant other [25, 26]. Among individuals undergoing social isolation and distancing during the COVID-19 pandemic in 2020, social support has been found to be inversely related to depression, anxiety, irritability, sleep quality, and loneliness, with higher levels of social support reducing the risk of depression and improving sleep quality [27]. Lastly, it has been shown in a 2020 study that social support builds resilience, a protective factor against depression, anxiety, and stress [28]. Therefore, given the protective effects of social support on psychological health, a supportive environment should be maintained in the classroom. Online learning must be perceived as an inclusive community and a safe space for peer-to-peer interactions [29]. This is echoed in another study in 2019 on depressed students who narrated their need to see themselves reflected on others [30]. Whether or not online learning currently implemented has successfully transitioned remains to be seen.

The effect of social media on students' mental health has been a topic of interest even before the pandemic [31, 32]. A systematic review published in 2020 found that social media use is responsible for aggravating mental health problems and that prominent risk factors for depression and anxiety include time spent, activity, and addiction to social media [31]. Another systematic review published in 2016 argues that the nature of online social networking use may be more important in influencing the symptoms of depression than the duration or frequency of the engagement—suggesting that social rumination and comparison are likely to be candidate mediators in the relationship between depression and social media [33]. However, their findings also suggest that the relationship between depression and online social networking is complex and necessitates further research to determine the impact of moderators and mediators that underly the positive and negative impact of online social networking on wellbeing [33].

Despite existing studies already painting a picture of the psychological effects of COVID-19 in the Philippines, to our knowledge, there are still no local studies contextualized to college students living in different regions of the country. Therefore, it is crucial to elicit the reasons and risk factors for depression, stress, and anxiety and determine the potential impact that

online learning and social media use may have on the mental health of the said population. In turn, the findings would allow the creation of more context-specific and regionalized interventions that can promote mental wellness during the COVID-19 pandemic.

## Materials and methods

### Study aims

The study's general objective is to assess the mental health status of college students and determine the different factors that influenced them during the COVID-19 pandemic. Specifically, it aims:

1. To describe the study population's characteristics, categorized by their mental health status, which includes depression, anxiety, and stress.

2. To determine the prevalence and risk factors of depression, anxiety, and stress among college students during the COVID-19 pandemic, quarantine, and subsequent shift to online learning.

3. To estimate the effect of social media use on depression, anxiety, stress, and coping strategies towards stress among college students and examine whether participant characteristics modified these associations.

4. To estimate the effect of online learning shift on depression, anxiety, stress, and coping strategies towards stress among college students and examine whether participant characteristics modified these associations.

5. To determine the facilitators or stressors among college students that modified their mental health status during the COVID-19 pandemic, quarantine, and subsequent shift to online learning.

### Study design

A mixed-method study design will be used to address the study's objectives, which will include Key Informant Interviews (KIIs) and an online survey. During the quarantine period of the COVID-19 pandemic in the Philippines from April to November 2021, the study shall occur with the population amid community quarantine and an abrupt transition to online classes. Since this is the Philippines' first study that will look at the prevalence of depression, anxiety, and stress among college students during the COVID-19 pandemic, quarantine, and subsequent shift to online learning, the online survey will be utilized for the quantitative part of the study design. For the qualitative component of the study design, KIIs will determine facilitators or stressors among college students that modified their mental health status during the quarantine period.

### Study population

The Red Cross Youth (RCY), one of the Philippine Red Cross's significant services, is a network of youth volunteers that spans the entire country, having active members in Luzon, Visayas, and Mindanao. The group is clustered into different age ranges, with the College Red Cross Youth (18–25 years old) being the study's population of interest. The RCY has over 26,060 students spread across 20 chapters located all over the country's three major island groups. The RCY is heterogeneously composed, with some members classified as college students and some as out-of-school youth. Given their nationwide scope, disseminating

information from the national to the local level is already in place; this is done primarily through email, social media platforms, and text blasts. The research team will leverage these platforms to distribute the online survey questionnaire.

In addition, the online survey will also be open to non-members of the RCY. It will be disseminated through social media and engagements with different university administrators in the country. Stratified random sampling will be done for the KIIs. The KII participants will be equally coming from the country's four (4) primary areas: 5–10 each from the national capital region (NCR), Luzon, Visayas, and Mindanao, including members and non-members of the RCY.

## Inclusion and exclusion criteria

The inclusion criteria for the online survey will include those who are 18–25 years old, currently enrolled in a university, can provide consent for the study, and are proficient in English or Filipino. The exclusion criteria will consist of those enrolled in graduate-level programs (e.g., MD, JD, Master's, Doctorate), out-of-school youth, and those whose current curricula involve going on duty (e.g., MDs, nursing students, allied medical professions, etc.). The inclusion criteria for the KIIs will include online survey participants who are 18–25 years old, can provide consent for the study, are proficient in English or Filipino, and have access to the internet.

## Sample size

A continuity correction method developed by Fleiss et al. (2013) was used to calculate the sample size needed [34]. For a two-sided confidence level of 95%, with 80% power and the least extreme odds ratio to be detected at 1.4, the computed sample size was 1890. With an adjustment for an estimated response rate of 90%, the total sample size needed for the study was 2,100. To achieve saturation for the qualitative part of the study, 20 to 40 participants will be randomly sampled for the KIIs using the respondents who participated in the online survey [35].

## Study procedure

**Self-Administered questionnaire.**   The study will involve creating, testing, and distributing a self-administered questionnaire (SAQ). All eligible study participants will answer the SAQ on socio-demographic factors such as age, sex, gender, sexual orientation, residence, household income, socioeconomic status, smoking status, family history of mental health, and COVID-19 sickness of immediate family members or friends. The two validated survey tools, Depression, Anxiety, and Stress Scale (DASS-21) and Brief-COPE, will be used for the mental health outcome assessment [36–39]. The DASS-21 will measure the negative emotional states of depression, anxiety, and stress [40], while the Brief-COPE will measure the students' coping strategies [41].

For the exposure assessment of the students to social media and shift to online learning, the total time spent on social media (TSSM) per day will be ascertained by querying the participants to provide an estimated time spent daily on social media during and after their online classes. In addition, students will be asked to report their use of the eight commonly used social media sites identified at the start of the study. These sites include Facebook, Twitter, Instagram, LinkedIn, Pinterest, TikTok, YouTube, and social messaging sites Viber/WhatsApp and Facebook Messenger with response choices coded as "(1) never," "(2) less often," "(3) every few weeks," "(4) a few times a week," and "(5) daily" [42–44]. Furthermore, a global frequency score will be calculated by adding the response scores from the eight social media sites. The

global frequency score will be used as an additional exposure marker of students to social media [45]. The shift to online learning will be assessed using questions that will determine the participants' satisfaction with online learning. This assessment is comprised of 8 items in which participants will be asked to respond on a 5-point Likert scale ranging from 'strongly disagree' to 'strongly agree.'

The online survey will be virtually distributed in English using the Qualtrics XM™ platform. Informed consent detailing the purpose, risks, benefits, methods, psychological referrals, and other ethical considerations will be included before the participants are allowed to answer the survey. Before administering the online survey, the SAQ shall undergo pilot testing among twenty (20) college students not involved with the study. It aims to measure total test-taking time, respondent satisfaction, and understandability of questions. The survey shall be edited according to the pilot test participant's responses. Moreover, according to the Philippines' Data Privacy Act, all the answers will be accessible and used only for research purposes.

**Key informant interviews.** The research team shall develop the KII concept note, focusing on the extraneous factors affecting the student's mental health, behaviors, and coping mechanism. Some salient topics will include current stressors (e.g., personal, academic, social), emotional reactions to these stressors, and how they wish to receive support in response to these stressors. The KII will be facilitated by a certified psychologist/psychiatrist/social scientist and research assistants using various online video conferencing software such as Google Meet, Skype, or Zoom. All the KIIs will be recorded and transcribed for analysis. Furthermore, there will be a debriefing session post-KII to address the psychological needs of the participants. Fig 1 presents the diagrammatic flowchart of the study.

## Data analyses

**Quantitative data.** Descriptive statistics will be calculated, including the prevalence of mental health outcomes such as depression, anxiety, stress, and coping strategies. In addition, correlation coefficients will be estimated to assess the relations among the different mental health outcomes, covariates, and possible risk factors.

Associations between mental health outcomes and possible risk factors will be estimated using generalized linear models, a standard method for analyzing data in cross-sectional studies. Depending on how rare or common the mental health outcomes are, generalized linear models with either a Poisson distribution and log link function with a robust variance estimator or a Binomial distribution and logit link function will be used to estimate either the adjusted prevalence ratios (PRs) or odds ratios (ORs) with 95% confidence intervals (CIs), respectively [46–49]. Separate single-mental health outcome models will be evaluated, and the models will consider the general form:

$$E[Y_i] = n_0 + n_1 X_{1i} + n_2 X_{2i} + \cdots + n_r X_{ri} \qquad [1]$$

where $Y_i$ will be the mental health outcome (depression, anxiety, stress, and coping strategy) status of subject $i$ and covariates for subject $i$ will be denoted by $X_{1i}$ to $X_{ri}$ as the possible exposure risk factors (i.e., social media use and shift to online learning) and confounding factors (i.e., age, sex, gender, smoking status, family income, etc.). In addition, we will control for the covariates chosen *a priori* as potentially important predictors of mental health outcomes in all the models.

Several study characteristics as effect modifiers will also be assessed, including sex, gender, sexual orientation, family income, smoking status, family history of mental health, and Covid-19. We will include interaction terms between the dichotomized modifier variable and markers of social media use (total TSSM and global frequency score) and shift to online learning in

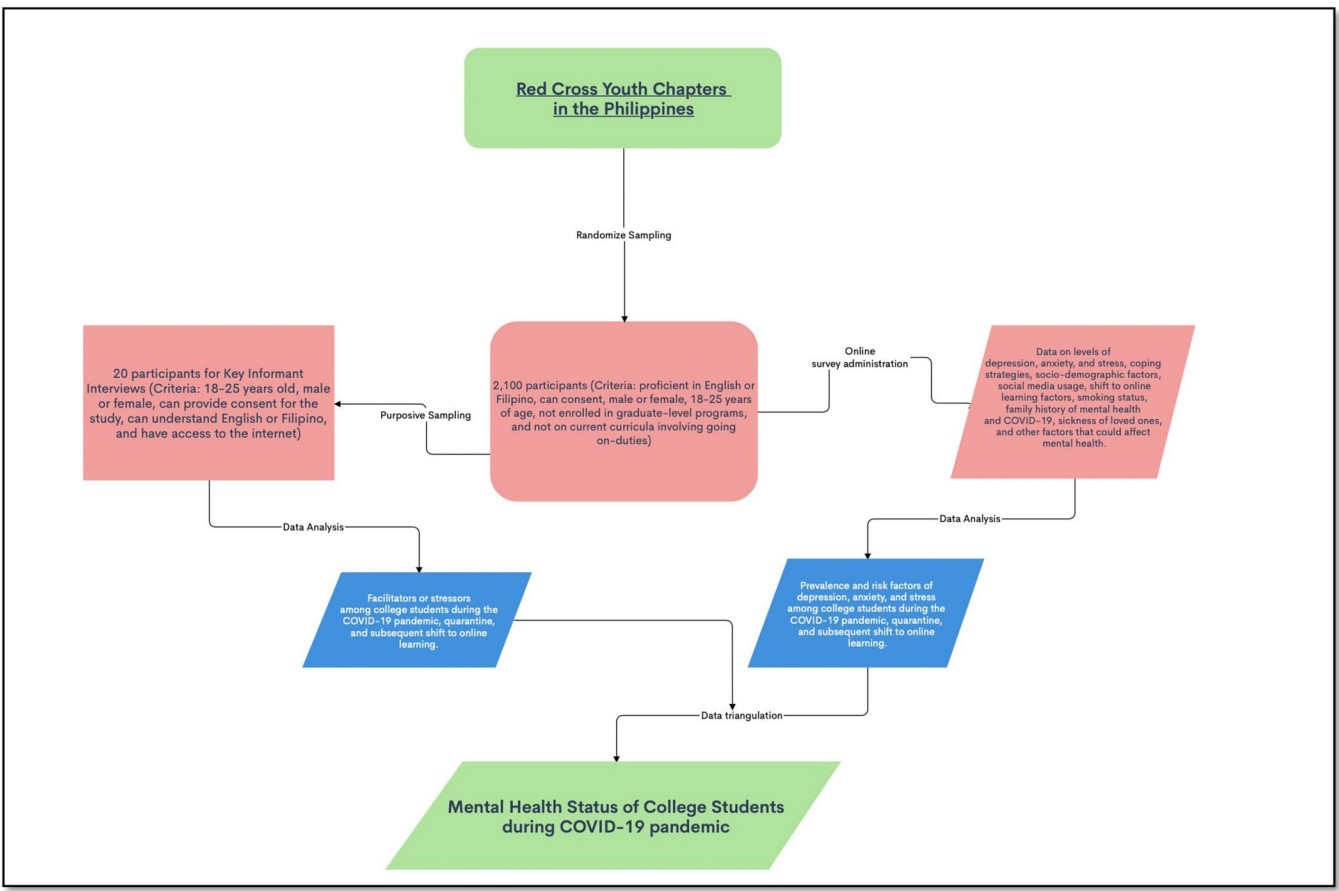

**Fig 1. Flow chart of the assessment of mental health status of college students during COVID-19 pandemic.**

the models. The significance of the interaction terms will be evaluated using the likelihood ratio test. All the regression analyses will be done in R (http://www.r-project.org). P values ≤ 0.05 will be considered statistically significant.

**Qualitative data.**    After transcribing the interviews, the data transcripts will be analyzed using NVivo 1.4.1 software [50] by three research team members independently using the inductive logic approach in thematic analysis: familiarizing with the data, generating initial codes, searching for themes, reviewing the themes, defining and naming the themes, and producing the report [51]. Data familiarization will consist of reading and re-reading the data while noting initial ideas. Additionally, coding interesting features of the data will follow systematically across the entire dataset while collating data relevant to each code. Moreover, the open coding of the data will be performed to describe the data into concepts and themes, which will be further categorized to identify distinct concepts and themes [52].

The three researchers will discuss the results of their thematic analyses. They will compare and contrast the three analyses in order to come up with a thematic map. The final thematic map of the analysis will be generated after checking if the identified themes work in relation to the extracts and the entire dataset. In addition, the selection of clear, persuasive extract examples that will connect the analysis to the research question and literature will be reviewed before producing a scholarly report of the analysis. Additionally, the themes and sub-themes generated will be assessed and discussed in relevance to the study's objectives. Furthermore,

the gathering and analyzing of the data will continue until saturation is reached. Finally, pseudonyms will be used to present quotes from qualitative data.

**Data triangulation.** Data triangulation using the two different data sources will be conducted to examine the various aspects of the research and will be compared for convergence. This part of the analysis will require listing all the relevant topics or findings from each component of the study and considering where each method's results converge, offer complementary information on the same issue, or appear to contradict each other. It is crucial to explicitly look for disagreements between findings from different data collection methods because exploration of any apparent inter-method discrepancy may lead to a better understanding of the research question [53, 54].

**Data management plan.** The Project Leader will be responsible for overall quality assurance, with research associates and assistants undertaking specific activities to ensure quality control. Quality will be assured through routine monitoring by the Project Leader and periodic cross-checks against the protocols by the research assistants. Transcribed KIIs and the online survey questionnaire will be used for recording data for each participant in the study. The project leader will be responsible for ensuring the accuracy, completeness, legibility, and timeliness of the data captured in all the forms. Data captured from the online survey or KIIs should be consistent, clarified, and corrected. Each participant will have complete source documentation of records. Study staff will prepare appropriate source documents and make them available to the Project Leader upon request for review. In addition, study staff will extract all data collected in the KII notes or survey forms. These data will be secured and kept in a place accessible to the Project Leader. Data entry and cleaning will be conducted, and final data cleaning, data freezing, and data analysis will be performed. Key informant interviews will always involve two researchers. Where appropriate, quality control for the qualitative data collection will be assured through refresher KII training during research design workshops. The Project Leader will check through each transcript for consistency with agreed standards. Where translations are undertaken, the quality will be assured by one other researcher fluent in that language checking against the original recording or notes.

## Ethics and dissemination

**Ethics approval.** The study shall abide by the Principles of the Declaration of Helsinki (2013). It will be conducted along with the Guidelines of the International Conference on Harmonization-Good Clinical Practice (ICH-GCP), E6 (R2), and other ICH-GCP 6 (as amended); National Ethical Guidelines for Health and Health-Related Research (NEGHHRR) of 2017. This protocol has been approved by the University of the Philippines Manila Research Ethics Board (UPMREB 2021-099-01 dated March 25, 2021).

The main concerns for ethics were consent, data privacy, and subject confidentiality. The risks, benefits, and conflicts of interest are discussed in this section from an ethical standpoint.

**Recruitment.** The participants will be recruited to answer the online SAQ voluntarily. The recruitment of participants for the KIIs will be chosen through stratified random sampling using a list of those who answered the online SAQ; this will minimize the risk of sampling bias. In addition, none of the participants in the study will have prior contact or association with the researchers. Moreover, power dynamics will not be contacted to recruit respondents. The research objectives, methods, risks, benefits, voluntary participation, withdrawal, and respondents' rights will be discussed with the respondents in the consent form before KII.

**Consent.** Informed consent will be signified by the potential respondent ticking a box in the online informed consent form and the voluntary participation of the potential respondent

to the study after a thorough discussion of the research details. The participant's consent is voluntary and may be recanted by the participant any time s/he chooses.

**Data privacy.**   All digital data will be stored in a cloud drive accessible only to the researchers. Subject confidentiality will be upheld through the assignment of control numbers and not requiring participants to divulge the name, address, and other identifying factors not necessary for analysis.

**Compensation.**   No monetary compensation will be given to the participants, but several tokens will be raffled to all the participants who answered the online survey and did the KIIs.

**Risks.**   This research will pose risks to data privacy, as discussed and addressed above. In addition, there will be a risk of social exclusion should data leaks arise due to the stigma against mental health. This risk will be mitigated by properly executing the data collection and analysis plan, excluding personal details and tight data privacy measures. Moreover, there is a risk of psychological distress among the participants due to the sensitive information. This risk will be addressed by subjecting the SAQ and the KII guidelines to the project team's psychiatrist's approval, ensuring proper communication with the participants. The KII will also be facilitated by registered clinical psychologists/psychiatrists/social scientists to ensure the participants' appropriate handling; there will be a briefing and debriefing of the participants before and after the KII proper.

**Benefits.**   Participation in this study will entail health education and a voluntary referral to a study-affiliated psychiatrist, discussed in previous sections. Moreover, this would contribute to modifications in targeted mental-health campaigns for the 18–25 age group. Summarized findings and recommendations will be channeled to stakeholders for their perusal.

**Dissemination.**   The results will be actively disseminated through conference presentations, peer-reviewed journals, social media, print and broadcast media, and various stakeholder activities.

## Discussion

This study protocol rationalizes the examination of the mental health of the college students in the Philippines during the COVID-19 pandemic as the traditional face-to-face classes transitioned to online and modular classes. The pandemic that started in March 2020 is now stretching for more than a year in which prolonged lockdown brings people to experience social isolation and disruption of everyday lifestyle. There is an urgent need to study the psychosocial aspects, particularly those populations that are vulnerable to mental health instability. In the Philippines, where community quarantine is still being imposed across the country, college students face several challenges amidst this pandemic. The pandemic continues to escalate, which may lead to fear and a spectrum of psychological consequences. Universities and colleges play an essential role in supporting college students in their academic, safety, and social needs. The courses of activities implemented by the different universities and colleges may significantly affect their mental well-being status. Our study is particularly interested in the effect of online classes on college students nationwide during the pandemic. The study will estimate this effect on their mental wellbeing since this abrupt transition can lead to depression, stress, or anxiety for some students due to insufficient time to adjust to the new learning environment. The role of social media is also an important exposure to some college students [55, 56]. Social media exposure to COVID-19 may be considered a contributing factor to college students' mental well-being, particularly their stress, depression, and anxiety [57, 58]. Despite these known facts, little is known about the effect of transitioning to online learning and social media exposure on the mental health of college students during the COVID-19 pandemic in the Philippines. To our knowledge, this is the first study in the Philippines that will use a

mixed-method study design to examine the mental health of college students in the entire country. The online survey is a powerful platform to employ our methods.

Additionally, our study will also utilize a qualitative assessment of the college students, which may give significant insights or findings of the experiences of the college students during these trying times that cannot be captured on our online survey. The thematic findings or narratives from the qualitative part of our study will be triangulated with the quantitative analysis for a more robust synthesis. The results will be used to draw conclusions about the mental health status among college students during the pandemic in the country, which will eventually be used to implement key interventions if deemed necessary. A cross-sectional study design for the online survey is one of our study's limitations in which contrasts will be mainly between participants at a given point of time. In addition, bias arising from residual or unmeasured confounding factors cannot be ruled out.

The COVID-19 pandemic and its accompanying effects will persistently affect the mental wellbeing of college students. Mental health services must be delivered to combat mental instability. In addition, universities and colleges should create an environment that will foster mental health awareness among Filipino college students. The results of our study will tailor the possible coping strategies to meet the specific needs of college students nationwide, thereby promoting psychological resilience.

## Acknowledgments

The researchers would like to extend their gratitude to the executives of the Philippine Red Cross, notably Senator Richard J. Gordon (Chairman), Ms. Elizabeth S. Zavalla (Secretary-General), and Ms. Maria Theresa S. Bongiad (Manager, Red Cross Youth), for making this project a reality. We also would like to thank all Red Cross Youth Chapters in the Philippines for helping in the pre-implementation stage of the project.

## Author Contributions

**Conceptualization:** Emmanuel S. Baja.

**Funding acquisition:** Leonard Thomas S. Lim, J. Rem C. Dela Cruz, Frances Dominique V. Ho, Marcela S. Rodolfo, Emmanuel S. Baja.

**Methodology:** Zypher Jude G. Regencia, Josefina Ly-Uson.

**Project administration:** Emmanuel S. Baja.

**Supervision:** Zypher Jude G. Regencia, Emmanuel S. Baja.

**Visualization:** Zypher Jude G. Regencia.

**Writing – original draft:** Leonard Thomas S. Lim, Zypher Jude G. Regencia, J. Rem C. Dela Cruz, Frances Dominique V. Ho, Marcela S. Rodolfo, Emmanuel S. Baja.

**Writing – review & editing:** Zypher Jude G. Regencia, Emmanuel S. Baja.

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
