## [Decision Letter · Decision Letter 0]

5 Nov 2021

PONE-D-21-17998Assessing the Effect of the COVID-19 Pandemic, Shift to Online Learning, and Social Media Use on Mental Health Among College Students in the Philippines: A Mixed-Method Study ProtocolPLOS ONE

Dear Dr. Baja,

Thank you for submitting your manuscript to PLOS ONE. After careful consideration, we feel that it has merit but does not fully meet PLOS ONE’s publication criteria as it currently stands. Therefore, we invite you to submit a revised version of the manuscript that addresses the points raised during the review process. Please address all comments from reviewers 1 and 2. Please disregard the comments from reviewers 3 and 4 to the effect that protocols shouldn't be published; PLOS ONE considers study protocols for publication (https://journals.plos.org/plosone/s/what-we-publish#loc-study-protocols).

We look forward to receiving your revised manuscript.

Kind regards,

Yann Benetreau, PhD

Senior Editor

PLOS ONE

Journal Requirements:

“The research received a grant from the American Red Cross through the Philippine Red Cross and Red Cross Youth.”

“This project is being supported by the American Red Cross through the Philippine Red Cross and Red Cross Youth. The funder will not have a role in the study design, data collection and analysis, decision to publish, or preparation of the manuscript.”

3. Please include a caption for figure 1.

Additional Editor Comments (if provided):

Dear Author

The manuscript is deemed to be not suitable for accepting for corrections or publications as it never meets the basic quality of an SSCI journal. Please refer to the comments from the reviewers.

Reviewers' comments:

Reviewer's Responses to Questions

**Comments to the Author**

1. Does the manuscript provide a valid rationale for the proposed study, with clearly identified and justified research questions?

Reviewer #1: Yes

Reviewer #2: Partly

Reviewer #3: Yes

Reviewer #4: No

2. Is the protocol technically sound and planned in a manner that will lead to a meaningful outcome and allow testing the stated hypotheses?

Reviewer #1: Yes

Reviewer #2: Partly

Reviewer #3: Partly

Reviewer #4: Partly

3. Is the methodology feasible and described in sufficient detail to allow the work to be replicable?

Reviewer #1: Yes

Reviewer #2: No

Reviewer #3: Yes

Reviewer #4: Yes

4. Have the authors described where all data underlying the findings will be made available when the study is complete?

Reviewer #1: Yes

Reviewer #2: No

Reviewer #3: No

Reviewer #4: No

5. Is the manuscript presented in an intelligible fashion and written in standard English?

Reviewer #1: Yes

Reviewer #2: No

Reviewer #3: Yes

Reviewer #4: Yes

6. Review Comments to the Author

You may also provide optional suggestions and comments to authors that they might find helpful in planning their study.

Reviewer #1: This study protocol aims to access the psychological effects among college students (18-25 years old) in the Philippines from the global pandemic, COVID-19, shift to online learning, and social media usage. The objectives of the study protocol address using a mixed-method study design that utilizes the quantitative and qualitative components. For the quantitative analysis, the authors propose sending an online self-administered questionnaire to the eligible participants to answer on socio-demographic factors. Based on the information provided, mental health outcomes will be assessed using two validated survey tools, Depression, Anxiety, and Stress Scale (DASS-21) and Brief-COPE. Additionally, the authors propose estimating the association between mental health outcomes and possible risk factors by using generalized linear models. Key informant interviews, a part of the qualitative component that addresses the stressors affecting the student’s mental health and behavior during the quarantine period. Finally, the authors suggest evaluating the data from quantitative and qualitative sources by using Data triangulation, which analyzes multiple sources of data to enhance the credibility of a research study.

The careful methodology provided in this study protocol will allow other researchers to apply this design to their studies. The validation of this study should provide a roadmap to study the effect of the pandemic on students in other countries.

The design of the study is very detailed for the most part. The authors have provided the necessary information about how the study population would be recruited and provided a justification for the sample size (quantitative data) that would be included in the study by providing relevant power calculations. However, for the qualitative study increasing the number of participants from different areas of the country would improve the quality of the outcome. Also, I would like to ask if any of the authors are Psychologists? If not, please acknowledge the Psychologists if the authors received any help in designing the study.

Also, the inclusion and exclusion criteria of the study population could be explained in more detail. It is not clear if only currently enrolled students would be included in the study. It would be important to justify the exclusion criteria.

Acknowledging similar studies (Copeland et al., 2021 and Fawaz et al., 2021) would help readers with a greater context. I would like to suggest the authors to cite the peer-reviewed version of the article titled “Barriers to online learning in the time of COVID-19: A national survey of medical students in the Philippines”. Also, I would like to ask the authors to change the references according to the journal requirements and have a uniform style.

Reviewer #2: Thank you for the opportunity to review the study protocol. The protocol is for a mixed-methods study looking at the impact of COVID19 and the subsequent shift to quarantine (stay at home), the implementation of online learning formats, and social media use on college students mental health. The study will provide insight into factors impacting mental health of college students. There are some issues with the study protocol that should be addressed.

Introduction

Paragraph 2.

• If you refer to SARS-CoV-2 as COVID-19, please include swine flu when referring to H1N1.

Paragraph 3.

• It is not clear which pandemic the authors are referring to.

• Include reference examples for the first sentence. P

• rovide examples of how infection and death have "adversely affected" mental health.

• How many people responded to the survey measuring the psychological effects of COVID19 in the Philippines?

Paragraph 4.

• Provide references for the first three sentences.

• Provide details of the studies you reference.

• Final sentence is conflating being infected by COVID19 and suffering from mood and substance disorders - please clarify exactly what is meant.

Paragraph 5.

• University students are not generally accepted as a vulnerable population. Please provide a reference that supports this statement.

• Second sentence - provide the references for the multiple studies.

• Are Chinese university students similar to Filipino college students? Surely there are other studies from other countries that can be included here. Or is the social, cultural, and political situation similar between the Philippines and China?

Paragraph 6.

• Second sentence is not clear. Do you mean that academic performance is associated with student mental health? If so, just say that.

• What is "this" in the sentence: "Online learning poses multiple challenges to this".

• Provide an example of "Students’ various social support systems" that have to adapt.

• In this sentence: "These challenges are alarming because social support has been noted as a critical aspect of mediating acute distress disorder" it's not clear if the statement refers to students or some other population.

• References are needed for the following sentences: "In addition, loneliness has been rising for the past six years amongst this vulnerable demographic. One study showed that being a student is a risk factor for loneliness, exacerbated during the pandemic." Furthermore, please provide greater clarity around the population being discussed.

• The following sentence does not follow the logic from the preceding sentences: "Therefore, online learning must be perceived as an inclusive community and a safe space for peer-to-peer interactions (18)."

Paragraph 7.

• The following sentence needs a reference: "One research recommends clear and focused design elements on accommodating students living with depression."

• The argument presented in this paragraph is not clear.

Paragraph 8

• The first three sentences need references.

• The argument for examining the effect of social media on students mental health is weak and is only presented in the second to last paragraph.

Paragraph 9

• There is no inclusion of social media in this paragraph.

• This paragraph should present a strong argument for the study, including all the factors that are to be included in the study.

Materials and methods

Study aim –

• Aims 1 and 2 are very similar. Aim 1 suggests you are going to describe the sample according to the categories of mental health. This is quite unusual and makes me think that maybe 'stratified is not the correct word for this aim. Perhaps what is meant is that the study aims to describe the characteristics of the sample population including mental health (e.g., depression, anxiety, stress)

• Aim 2 it is not clear if the determination of prevalence is before or post, the subsequent shift to online learning. Please clarify.

• Aim 3. It is not clear what the aim is. Please simplify. It might require breaking this one aim up into 2 or 3.

• Aim 4 Is similar to the last aspect of aim 3.

• Furthermore, the phrase "during the COVID-19 pandemic, quarantine, and subsequent shift to online learning." Is confusing - COVID 19 is current, it's not clear whether all or some of the students are in quarantine, and presumably, they have shifted to online learning (past). Please clarify the state in which the study will be conducted.

Design,

• When is the quarantine period. Please provide dates?

• Population RCY seems like a great way to recruit participants. However, this population may not reflect all college students as RCY are volunteers. Students who volunteer may have different values and attitudes towards mental health, and social justice and adapting to change. The authors must account for this in their study and ensure there are no differences between their RCY participants and the non-member RCY participants on critical factors (e.g., mental health etc).

• It is not clear how random sampling for the KII will be achieved. The reported sampling method reads more like stratified sampling.

Inclusion exclusion criteria

• It is not clear why those who identify as non-binary genders are not included. Why is gender an inclusion/ exclusion criterion?

• The sample size calculation - how was the number of KIIs determined? how data saturation will be determined

Study procedure

• It's not clear how the demographic factors will be collected.

• Variables are not clear (e.g., sickness of loved ones) - do you mean family members? friends? pets? do you mean chronic illness or acute illness?

• What other factors that could affect mental health are you going to measure?

• This statement is not accurate: "The DASS-21 will measure the prevalence of depression, anxiety, and stress-related issues affecting daily life (28)" please correct to more accurately describe the DASS.

• It is not clear how this variable will be measured: "the total time spent on social media (TSSM) per day will be ascertained by querying the participants to provide an estimated time spent daily on social media during and after their online classes." is it the total time in one day, or only during and after class. Why not all day? or why is it only during and after of interest? The assumption is that they will increase their use of social media, but what if their use of social media is the same or less than before the shift to online learning?

• Regarding the KII, it may be country-specific, but it's not clear how social scientists and research assistants facilitation of interviews will be the same quality as psychologists and psychiatrists. How will the difference in skills in interviewing be overcome?

• Will the results of the survey be used to develop interview questions?

Data Analysis

• Given that all the variables are known, the quantitative analysis could be clearer with examples of what the authors mean by 'covariate' and 'possible risk factors.

• Will the analyses be explorational? the literature review implies that some hypotheses may be developed. If so, the analyses should be designed to test those hypotheses.

• How will p-values be adjusted to account for the multiple analyse?

Qualitative data

• Will the themes be developed independently by researchers? how many researchers will be involved in the coding? It is not clear from the description how will triangulation be established. Will multiple authors do the coding of the interview transcripts - independently? Will the results of the survey be used to inform the coding of the interviews?

Ethics and dissemination

• It is still not clear how random sampling will be achieved by the authors for recruitment for KII.

• What will the researchers do if a participants response to the DASS indicates they have clinical levels of Depression, Anxiety or Stress?

Discussion

• References are needed throughout the discussion. For example, "The role of social media is also an important exposure to some college students. Social media exposure to COVID-19 may be considered a contributing factor to college students’ mental well-being, particularly their stress, depression, and anxiety."

Reviewer #3: The title of the protocol is timely and well presented. However, I don't agree on publishing protocols for cross-sectional studies. However, the results of this protocol are expected to add great value for public health.

Reviewer #4: This is just a proposal stage. Some part of the methods section is not well defined. Without any results, it is not suitable for a scientific publication yet.

7. PLOS authors have the option to publish the peer review history of their article (what does this mean?). If published, this will include your full peer review and any attached files.

Reviewer #1: No

Reviewer #2: No

Reviewer #3: No

Reviewer #4: No

---

## [Author Response · Author response to Decision Letter 0]

16 Jan 2022

Reviewer No. 1:

This study protocol aims to assess the psychological effects among college students (18-25 years old) in the Philippines from the global pandemic, COVID-19, shift to online learning, and social media usage. The objectives of the study protocol address using a mixed-method study design that utilizes the quantitative and qualitative components. For the quantitative analysis, the authors propose sending an online self-administered questionnaire to the eligible participants to answer on socio-demographic factors. Based on the information provided, mental health outcomes will be assessed using two validated survey tools, Depression, Anxiety, and Stress Scale (DASS-21) and Brief-COPE. Additionally, the authors propose estimating the association between mental health outcomes and possible risk factors by using generalized linear models. Key informant interviews, a part of the qualitative component that addresses the stressors affecting the student’s mental health and behavior during the quarantine period. Finally, the authors suggest evaluating the data from quantitative and qualitative sources by using data triangulation, which analyzes multiple sources of data to enhance the credibility of a research study.

The careful methodology provided in this study protocol will allow other researchers to apply this design to their studies. The validation of this study should provide a roadmap to study the effect of the pandemic on students in other countries.

The design of the study is very detailed for the most part. The authors have provided the necessary information about how the study population would be recruited and provided a justification for the sample size (quantitative data) that would be included in the study by providing relevant power calculations. However, for the qualitative study, increasing the number of participants from different areas of the country would improve the quality of the outcome. Also, I would like to ask if any of the authors are Psychologists? If not, please acknowledge the Psychologists if the authors received any help in designing the study.

Response: One of the authors is a senior consultant psychiatrist, Dr. Josefina T. Ly-Uson. She helped in the design of the study. 

Also, the inclusion and exclusion criteria of the study population could be explained in more detail. It is not clear if only currently enrolled students would be included in the study. It would be important to justify the exclusion criteria.

Response: The inclusion criteria now included “currently enrolled college students.” The exclusion criteria included those students in graduate-level programs and those whose current curricula involve going on duty. We purposely chose to exclude those students because they have a different set of schooling conditions compared to the rest of the regular college students. 

Acknowledging similar studies (Copeland et al., 2021 and Fawaz et al., 2021) would help readers with a greater context. I would like to suggest the authors cite the peer-reviewed version of the article titled “Barriers to online learning in the time of COVID-19: A national survey of medical students in the Philippines”. Also, I would like to ask the authors to change the references according to the journal requirements and have a uniform style.

Response: Thank you for the suggestion. This comment is noted, and revisions have been made. 

Reviewer No. 2

Thank you for the opportunity to review the study protocol. The protocol is for a mixed-methods study looking at the impact of COVID19 and the subsequent shift to quarantine (stay at home), the implementation of online learning formats, and social media use on college students’ mental health. The study will provide insight into factors impacting mental health of college students. There are some issues with the study protocol that should be addressed.

Introduction

Paragraph 2.

• If you refer to SARS-CoV-2 as COVID-19, please include swine flu when referring to H1N1.

Response: This is noted, and revisions have been made. 

Paragraph 3.

• It is not clear which pandemic the authors are referring to.

Response: This is noted, and revisions have been made. 

• Include reference examples for the first sentence.

Response: This is noted, and revisions have been made. 

• Provide examples of how infection and death have "adversely affected" mental health.

Response: This is noted, and revisions have been made. 

• How many people responded to the survey measuring the psychological effects of COVID19 in the Philippines?

Response: This is noted, and revisions have been made. 

Paragraph 4.

• Provide references for the first three sentences.

Response: This is noted, and revisions have been made. 

• Provide details of the studies you reference.

Response: This is noted, and revisions have been made. 

• Final sentence is conflating being infected by COVID19 and suffering from mood and substance disorders - please clarify exactly what is meant.

Response: This is noted, and revisions have been made. 

Paragraph 5.

• University students are not generally accepted as a vulnerable population. Please provide a reference that supports this statement.

Response: This is noted, and the statement was deleted.

• Second sentence - provide the references for the multiple studies.

Response: This is noted, and revisions have been made. 

• Are Chinese university students similar to Filipino college students? Surely there are other studies from other countries that can be included here. Or is the social, cultural, and political situation similar between the Philippines and China?

Response: This is noted, and revisions have been made. We cited prevalence rates from three different countries (USA, Malaysia, and China). 

Paragraph 6.

• Second sentence is not clear. Do you mean that academic performance is associated with student mental health? If so, just say that.

Response: This is noted, and revisions have been made. 

• What is "this" in the sentence: "Online learning poses multiple challenges to this".

Response: This is noted, and revisions have been made. 

• Provide an example of "Students’ various social support systems" that have to adapt.

Response: This is noted and revisions have been made. 

• In this sentence: "These challenges are alarming because social support has been noted as a critical aspect of mediating acute distress disorder" it's not clear if the statement refers to students or some other population.

Response: This is noted, and revisions have been made. 

• References are needed for the following sentences: "In addition, loneliness has been rising for the past six years amongst this vulnerable demographic. One study showed that being a student is a risk factor for loneliness, exacerbated during the pandemic." Furthermore, please provide greater clarity around the population being discussed.

Response: This is noted, and revisions have been made. 

• The following sentence does not follow the logic from the preceding sentences: "Therefore, online learning must be perceived as an inclusive community and a safe space for peer-to-peer interactions (18)."

Response: This is noted, and revisions have been made. 

Paragraph 7.

• The following sentence needs a reference: "One research recommends clear and focused design elements on accommodating students living with depression."

Response: This is noted, and revisions have been made. 

• The argument presented in this paragraph is not clear.

Response: This is noted, and revisions have been made. 

Paragraph 8

• The first three sentences need references.

Response: This is noted, and references have been added. 

• The argument for examining the effect of social media on students' mental health is weak and is only presented in the second to last paragraph.

Response: This is noted, and revisions have been made. 

Paragraph 9

• There is no inclusion of social media in this paragraph.

Response: This is noted, and revisions have been made. 

• This paragraph should present a strong argument for the study, including all the factors that are to be included in the study.

Response: This is noted, and revisions have been made. 

Materials and methods

Study aim

• Aims 1 and 2 are very similar. Aim 1 suggests you are going to describe the sample according to the categories of mental health. This is quite unusual and makes me think that maybe 'stratified' is not the correct word for this aim. Perhaps what is meant is that the study aims to describe the characteristics of the sample population including mental health (e.g., depression, anxiety, stress)

Response: This is noted. We replaced the word stratified with categorized. Aims 1 and 2 are not similar. The first aim will present the characteristics of the population, while the second aim will present the prevalence and risk factors.

• Aim 2 it is not clear if the determination of prevalence is before or post, the subsequent shift to online learning. Please clarify.

Response: Aim 2 will determine the prevalence after the subsequent shift to online learning. We will not determine the pre-shift to online learning prevalence because there was an abrupt transition to online learning in the Philippines, which limits our time frame to collect data before the shift to online learning. 

• Aim 3. It is not clear what the aim is. Please simplify. It might require breaking this one aim up into 2 or 3.

Response: Aim 3 was split into 2 aims. The first aim will be looking at the effect of social media use on markers of mental health (depression, anxiety, stress, and coping strategies towards stress). At the same time, the second aim will look at the effect of online learning shift on markers of mental health (depression, anxiety, stress, and coping strategies towards stress).

• Aim 4 Is similar to the last aspect of aim 3.

Response: Aim 4 is very different from Aim 3 because Aim 4 is the qualitative part of the study which will use key informant interviews to explore facilitators or stressors that modified the mental health status of the participants while Aim 3 is the quantitative part. 

• Furthermore, the phrase "during the COVID-19 pandemic, quarantine, and subsequent shift to online learning." Is confusing - COVID 19 is current, it's not clear whether all or some of the students are in quarantine, and presumably, they have shifted to online learning (past). Please clarify the state in which the study will be conducted.

Response: Currently, the Philippines is still in quarantine due to COVID-19 and all classes made a shift to online learning and no face-to-face classes are allowed. 

Design

• When is the quarantine period? Please provide dates?

Response: Currently, the Philippines is still in quarantine. We opted to use April to November 2021 as the study period. 

• Population RCY seems like a great way to recruit participants. However, this population may not reflect all college students as RCY are volunteers. Students who volunteer may have different values and attitudes towards mental health, and social justice and adapting to change. The authors must account for this in their study and ensure there are no differences between their RCY participants and the non-member RCY participants on critical factors (e.g., mental health etc).

Response: RCY is connected with the majority of the universities and colleges in the Philippines. RCY will spearhead the distribution of the questionnaires to universities and colleges in the Philippines. Therefore, non-RCY volunteers are also encouraged and recruited to participate in the study. We will account for the differences between RCY and non-RCY volunteers by adding an indicator variable in our statistical models. 

• It is not clear how random sampling for the KII will be achieved. The reported sampling method reads more like stratified sampling.

Response: Yes, it is a stratified random sampling using 4 major regions of the country. We have revised the section. 

Inclusion exclusion criteria

• It is not clear why those who identify as non-binary genders are not included. Why is gender an inclusion/ exclusion criterion?

Response: Gender is not an exclusion/inclusion criterion. Revisions have been made. 

• The sample size calculation - how was the number of KIIs determined? how data saturation will be determined

Response: The saturation will be achieved once almost all of the interview transcripts have been generating no new information. We added the reference Hagaman and Wutich as basis for the KII sample size. 

Study procedure

• It's not clear how the demographic factors will be collected.

Response: Demographic factors will be collected using the SAQ we developed. 

• Variables are not clear (e.g., sickness of loved ones) - do you mean family members? friends? pets? do you mean chronic illness or acute illness?

Response: This is noted and revisions have been made. 

• What other factors that could affect mental health are you going to measure?

Response: We deleted “other factors that could affect mental health”. 

• This statement is not accurate: "The DASS-21 will measure the prevalence of depression, anxiety, and stress-related issues affecting daily life (28)" please correct to more accurately describe the DASS.

Response: This is noted and revisions have been made. 

• It is not clear how this variable will be measured: "the total time spent on social media (TSSM) per day will be ascertained by querying the participants to provide an estimated time spent daily on social media during and after their online classes." is it the total time in one day, or only during and after class. Why not all day? or why is it only during and after interest? The assumption is that they will increase their use of social media, but what if their use of social media is the same or less than before the shift to online learning?

Response: We will measure the social media during and outside online class hours. TSSM will measure the total time spent per day on social media. We will not account for the TSSM before shifting to online learning. Our objective is to measure their TSSM during and outside of class hours that is happening in a quarantine period. 

• Regarding the KII, it may be country-specific, but it's not clear how social scientists and research assistants facilitation of interviews will be the same quality as psychologists and psychiatrists. How will the difference in skills in interviewing be overcome?

Response: Dr. Uson, our board-certified psychiatrist, trained the research assistants to facilitate KIIs. She will also be present during the KII. In the Philippines, training of the research assistants to facilitate KII is done in order for them to learn and be adequately trained for future research. Research assistants may someday be the research primary investigators. 

• Will the results of the survey be used to develop interview questions?

Response: The results of the survey will not be used to develop interview questions. But we will use the answers of the KII participants in the survey as a guide during the interviews of the KII participants. 

Data Analysis

• Given that all the variables are known, the quantitative analysis could be clearer with examples of what the authors mean by 'covariate' and 'possible risk factors.

Response: Yes, all the possible variables that will be used in the data analysis have been identified a priori as confounding factor covariates and risk factors. 

• Will the analyses be explorational? The literature review implies that some hypotheses may be developed. If so, the analyses should be designed to test those hypotheses.

Response: The analysis will not be explorational. Important possible risk factors (exposures) and covariates (confounding factors) were chosen a priori and will be used for the multivariable generalized linear models. 

• How will p-values be adjusted to account for the multiple analyses?

Response: We will not use a correction method to adjust the p-values, since our statistical analyses will be not that many to warrant an adjustment. Possible adjustments of p-values are for microarray datasets to correct the occurrence of false positives in the multiple analyses. 

Qualitative data

• Will the themes be developed independently by researchers? How many researchers will be involved in the coding? It is not clear from the description how triangulation will be established. Will multiple authors do the coding of the interview transcripts - independently? Will the results of the survey be used to inform the coding of the interviews?

Response: This is noted, and revisions have been made. The description of the triangulation was described in the “data triangulation” section and references have been updated to include the “comparison of datasets for convergence”. As mentioned earlier, the results of the survey will not be used to develop interview questions. But we will use their answers in the survey as a guide during the interview. 

Ethics and dissemination

• It is still not clear how random sampling will be achieved by the authors for recruitment for KII.

Response: From the pool of participants per area, using a software, we will randomly choose prospective KII participants and contact them if they are willing to participate. If they opted not to participate, we will randomly choose participants again from the pool of respondents per area. 

• What will the researchers do if a participant's response to the DASS indicates they have clinical levels of Depression, Anxiety or Stress?

Response: It is our ethical duty to assist the participants if they have been screened to have high clinical levels of depression, anxiety, or stress. We will refer them to a tertiary government hospital for further evaluation, or treatment if needed. 

Discussion

• References are needed throughout the discussion. For example, "The role of social media is also an important exposure to some college students. Social media exposure to COVID-19 may be considered a contributing factor to college students’ mental well-being, particularly their stress, depression, and anxiety."

Response: This is noted, and references have been added.

---

## [Decision Letter · Decision Letter 1]

15 Feb 2022

PONE-D-21-17998R1Assessing the Effect of the COVID-19 Pandemic, Shift to Online Learning, and Social Media Use on the Mental Health of College Students in the Philippines: A Mixed-Method Study ProtocolPLOS ONE

Dear Dr. Baja,

Thank you for submitting your manuscript to PLOS ONE. After careful consideration, we feel that it has merit but does not fully meet PLOS ONE’s publication criteria as it currently stands. Therefore, we invite you to submit a revised version of the manuscript that addresses the points raised during the review process.

 Please submit your revised manuscript by Mar 31 2022 11:59PM. If you will need more time than this to complete your revisions, please reply to this message or contact the journal office at plosone@plos.org. Please include the following items when submitting your revised manuscript:A rebuttal letter that responds to each point raised by the academic editor and reviewer(s). You should upload this letter as a separate file labeled 'Response to Reviewers'.A marked-up copy of your manuscript that highlights changes made to the original version. You should upload this as a separate file labeled 'Revised Manuscript with Track Changes'.An unmarked version of your revised paper without tracked changes. You should upload this as a separate file labeled 'Manuscript'.If applicable, we recommend that you deposit your laboratory protocols in protocols.io to enhance the reproducibility of your results. Protocols.io assigns your protocol its own identifier (DOI) so that it can be cited independently in the future. For instructions see: https://journals.plos.org/plosone/s/submission-guidelines#loc-laboratory-protocols. Additionally, PLOS ONE offers an option for publishing peer-reviewed Lab Protocol articles, which describe protocols hosted on protocols.io. Read more information on sharing protocols at https://plos.org/protocols?utm_medium=editorial-email&utm_source=authorletters&utm_campaign=protocols.

We look forward to receiving your revised manuscript.

Kind regards,

Jianhong Zhou

Associate Editor

PLOS ONE

Journal Requirements:

Reviewers' comments:

Reviewer's Responses to Questions

**Comments to the Author**

1. Does the manuscript provide a valid rationale for the proposed study, with clearly identified and justified research questions?

Reviewer #1: Yes

Reviewer #2: Yes

Reviewer #4: Partly

2. Is the protocol technically sound and planned in a manner that will lead to a meaningful outcome and allow testing the stated hypotheses?

Reviewer #1: Yes

Reviewer #2: Yes

Reviewer #4: Partly

3. Is the methodology feasible and described in sufficient detail to allow the work to be replicable?

Reviewer #1: Yes

Reviewer #2: Yes

Reviewer #4: Yes

4. Have the authors described where all data underlying the findings will be made available when the study is complete?

Reviewer #1: Yes

Reviewer #2: Yes

Reviewer #4: No

5. Is the manuscript presented in an intelligible fashion and written in standard English?

Reviewer #1: Yes

Reviewer #2: Yes

Reviewer #4: Yes

6. Review Comments to the Author

You may also provide optional suggestions and comments to authors that they might find helpful in planning their study.

Reviewer #1: Appreciate the authors' efforts, they addressed all the concerns adequately.

However, I suggest authors to cite the peer-reviewed version of the article instead of medRxiv.

Good luck with your publication.

Reviewer #2: Thank you for the opportunity to review this protocol. The authors have made considerable changes to the protocol in response to reviewer comments. There are just a couple of minor points to be addressed.

Paragraph 6 – please include the years that these studies were conducted. Also, as these studies are cross-sectional, it is not entirely accurate to say that there is a growing prevalence (which you could do if there were similar studies conducted years apart on the same population, or a longitudinal study. But these are separate populations and the years have not been presented to the reader). Perhaps just delete ‘growing’

Qualitative data – paragraph 1- The description of the qualitative data is missing some information. For example, at what point will the three qualitative analysts discuss the themes? Will the three analysts do all of the transcripts or will they do a sample to establish consistency before dividing the transcripts? Please confirm whether inductive or deductive logic to the coding approach.

Reviewer #4: -The study is proposed to explore the effect of social media use, online learning upon mental health. It is hard to differentiate wither the effect is due to social media or online learning. (aim 3 and aim 4).

- The study participants (inclusion criteria) also include out-of-school youth as they are including all RCY who agreed.

-

7. PLOS authors have the option to publish the peer review history of their article (what does this mean?). If published, this will include your full peer review and any attached files.

Reviewer #1: No

Reviewer #2: No

Reviewer #4: No

---

## [Author Response · Author response to Decision Letter 1]

23 Feb 2022

RESPONSE TO REVIEWERS

PONE-D-21-17998: Assessing the Effect of the COVID-19 Pandemic, Shift to Online Learning, and Social Media Use on Mental Health Among College Students in the Philippines: A Mixed-Method Study Protocol

Reviewer #1: Appreciate the authors' efforts, they addressed all the concerns adequately. However, I suggest authors cite the peer-reviewed version of the article instead of medRxiv.

Good luck with your publication.

Response: We have removed the medRxiv references. 

Reviewer #2: Thank you for the opportunity to review this protocol. The authors have made considerable changes to the protocol in response to reviewer comments. There are just a couple of minor points to be addressed.

Paragraph 6 – please include the years that these studies were conducted. Also, as these studies are cross-sectional, it is not entirely accurate to say that there is a growing prevalence (which you could do if there were similar studies conducted years apart on the same population, or a longitudinal study. But these are separate populations and the years have not been presented to the reader). Perhaps just delete ‘growing’

Response: We have deleted the word “growing.”

Qualitative data – paragraph 1- The description of the qualitative data is missing some information. For example, at what point will the three qualitative analysts discuss the themes? Will the three analysts do all of the transcripts or will they do a sample to establish consistency before dividing the transcripts? Please confirm whether inductive or deductive logic to the coding approach.

Response: The qualitative data description was added with information regarding the discussion of thematic analysis of the three researchers. To ensure consistency, KII training, including transcription and quality assurance, will be done with the research team members as detailed in the Data Management Plan Section. Moreover, an inductive logic approach to coding will be conducted. All of these are reflected in the revised manuscript. 

Reviewer #4: -The study is proposed to explore the effect of social media use, online learning on mental health. It is hard to differentiate whether the effect is due to social media or online learning. (Aim 3 and Aim 4).

Response: Generalized linear model using Poisson regression will be done to independently analyze the effect of social media usage or online learning on the participants' mental health. Thus, social media usage and online learning will be treated as two exposures independently. 

- The study participants (inclusion criteria) also include out-of-school youth as they are including all RCY who agreed.

Response: We have revised the inclusion and exclusion criteria. “Out-of-school youth” classification is now included in the exclusion criteria. Moreover, current enrollment in a university is part of the inclusion criteria.

---

## [Decision Letter · Decision Letter 2]

12 Apr 2022

Assessing the Effect of the COVID-19 Pandemic, Shift to Online Learning, and Social Media Use on the Mental Health of College Students in the Philippines: A Mixed-Method Study Protocol

PONE-D-21-17998R2

Dear Dr. Baja,

We’re pleased to inform you that your manuscript has been judged scientifically suitable for publication and will be formally accepted for publication once it meets all outstanding technical requirements.

Kind regards,

Elisa Panada

Staff Editor

PLOS ONE

Additional Editor Comments (optional):

Reviewers' comments:

Reviewer's Responses to Questions

**Comments to the Author**

1. Does the manuscript provide a valid rationale for the proposed study, with clearly identified and justified research questions?

Reviewer #1: Yes

Reviewer #2: Yes

2. Is the protocol technically sound and planned in a manner that will lead to a meaningful outcome and allow testing the stated hypotheses?

Reviewer #1: Yes

Reviewer #2: Yes

3. Is the methodology feasible and described in sufficient detail to allow the work to be replicable?

Reviewer #1: Yes

Reviewer #2: Yes

4. Have the authors described where all data underlying the findings will be made available when the study is complete?

Reviewer #1: Yes

Reviewer #2: Yes

5. Is the manuscript presented in an intelligible fashion and written in standard English?

Reviewer #1: Yes

Reviewer #2: Yes

6. Review Comments to the Author

You may also provide optional suggestions and comments to authors that they might find helpful in planning their study.

Reviewer #1: I am satisfied with the revisions that were made to the manuscript. I endorse this manuscript for publication.

Reviewer #2: The manuscript provides a valid rationale for the proposed study and the study is technically sound. The methodology is feasible and described in detail.

7. PLOS authors have the option to publish the peer review history of their article (what does this mean?). If published, this will include your full peer review and any attached files.

Reviewer #1: No

Reviewer #2: No

---

## [Editor Report · Acceptance letter]

21 Apr 2022

PONE-D-21-17998R2 

Assessing the Effect of the COVID-19 Pandemic, Shift to Online Learning, and Social Media Use on the Mental Health of College Students in the Philippines: A Mixed-Method Study Protocol 

Dear Dr. Baja:

I'm pleased to inform you that your manuscript has been deemed suitable for publication in PLOS ONE. Congratulations! Your manuscript is now with our production department. 

Kind regards, 

on behalf of

Dr Elisa Panada 

%CORR_ED_EDITOR_ROLE%

PLOS ONE